# Riboceine Rescues Auranofin-Induced Craniofacial Defects in Zebrafish

**DOI:** 10.3390/antiox10121964

**Published:** 2021-12-08

**Authors:** Megan Leask, Catherine Carleton, Bryony Leeke, Trent Newman, Joseph Antoun, Mauro Farella, Julia Horsfield

**Affiliations:** 1Department of Pathology, Dunedin School of Medicine, University of Otago, Dunedin 9016, New Zealand; megan.leask@otago.ac.nz (M.L.); catherine.r.carleton@gmail.com (C.C.); b.leeke@lms.mrc.ac.uk (B.L.); tacnewman@gmail.com (T.N.); 2Maurice Wilkins Centre for Molecular Biodiscovery, Private Bag 92019, The University of Auckland, Auckland 1010, New Zealand; joseph.antoun@otago.ac.nz (J.A.); mauro.farella@otago.ac.nz (M.F.); 3Department of Oral Sciences, Sir John Walsh Research Institute, University of Otago, Dunedin 9016, New Zealand; 4Genetics Otago Research Centre, University of Otago, Dunedin 9016, New Zealand

**Keywords:** craniofacial anomalies, oxidative stress, auranofin, Riboceine, antioxidant, oxidative stress, neural crest, zebrafish

## Abstract

Craniofacial abnormalities are a common group of congenital developmental disorders that can require intensive oral surgery as part of their treatment. Neural crest cells (NCCs) contribute to the facial structures; however, they are extremely sensitive to high levels of oxidative stress, which result in craniofacial abnormalities under perturbed developmental environments. The oxidative stress-inducing compound auranofin (AFN) disrupts craniofacial development in wildtype zebrafish embryos. Here, we tested whether the antioxidant Riboceine (RBC) rescues craniofacial defects arising from exposure to AFN. RBC rescued AFN-induced cellular apoptosis and distinct defects of the cranial cartilage in zebrafish larvae. Zebrafish embryos exposed to AFN have higher expression of antioxidant genes *gstp1* and *prxd1*, with RBC treatment partially rescuing these gene expression profiles. Our data suggest that antioxidants may have utility in preventing defects in the craniofacial cartilage owing to environmental or genetic risk, perhaps by enhancing cell survival.

## 1. Introduction

The prevalence of craniofacial abnormalities varies widely across population groups, geographical location, socioeconomic status and environmental exposures. The etiology of craniofacial abnormalities is multifactorial, with evidence that genetic and environmental factors contribute to risk [1,2,3]. Major environmental risk factors include exposure to maternal diabetes, ethanol, nicotine and recreational drugs during embryo development [4,5,6,7]. Teratogenicity to the embryo following these environmental exposures is likely due to increased reactive oxygen species (ROS) and redox imbalance [8]. ROS are essential for normal embryonic development; however, alterations to redox status can disrupt normal fetal development [9]. The neural crest cells (NCCs), which contribute to the craniofacial structures in early development [10], are particularly sensitive to exogenous oxidative stress due to their existing high oxidative state [6]. Alterations in the redox state during the development of NCCs can result in autophagy or apoptosis mediated by molecular pathways, such as DNA damage and p53 activation [6,11,12,13], and therefore, redox balance is critical in craniofacial development. Consistently, genetic variants associated with conditions with craniofacial phenotypes have been found in DNA damage repair genes [6,14,15] and are associated with increased oxidative stress [6,16,17]. For preventative treatment of craniofacial abnormalities, environmental and genetic risks could be modified by identifying compounds that reduce oxidative stress and subsequent cell death of the NCCs.

The proteins Thioredoxin (Trx) and Glutathione (GSH) are potent intracellular antioxidants that protect cells from oxidative stress by balancing cellular levels of reactive oxygen species [14,15]. Previously, we have shown that auranofin (AFN), which alters cellular redox status by inhibiting Thioredoxin Reductase (TrxR) [16], disrupts the development of craniofacial cartilage in zebrafish larvae and induces expression of GSH synthesis genes [17]. We hypothesize that exogenous compounds that promote GSH synthesis might ameliorate AFN-induced craniofacial abnormalities by reducing oxidative stress. Cysteine pro-drugs increase levels of GSH by providing cysteine for gamma-glutamylcysteine ligase the rate-limiting enzyme in glutathione biosynthesis [18]. Riboceine (d-ribose-l-cysteine) is a possible alternative to other more well-studied cysteine pro-drugs including NAC because of its lower toxicity due to slower release of l-cysteine and bioavailability [19,20,21]. Additionally, there are conflicting reports on whether NAC can rescue developmental defects in zebrafish induced by oxidative stress [22,23,24,25,26]. Here, we test if Riboceine (RBC), which is now readily available as an over-the-counter supplement, can counteract the craniofacial damage caused by oxidative stress resulting from AFN treatment.

## 2. Materials and Methods

### 2.1. Animal Husbandry

Adult zebrafish (wildtype AB line) were maintained under standard conditions [27]. The study conforms to the ARRIVE Guidelines. Briefly, zebrafish were housed in 3.5 L transparent polycarbonate tanks, housed in a Central Live Support System (Tecniplast^®^ Aquatic Solutions, West Chester, PA, USA) in the University of Otago Zebrafish Facility. Zebrafish were fed two dry feeds and one live feed of brine shrimp (Artemia salina) each day. For breeding, male:female pairs were placed into a breeding box (Tecniplast^®^ Aquatic Solutions, West Chester, PA, USA) and separated overnight. In the morning, the separation barrier between male and female zebrafish pairs was removed to allow spawning to occur. After breeding (up to 30 min) zebrafish embryos were collected with a sieve, transferred into a petri dish (100 mm × 15 mm, ~30 embryos per dish) containing E3 embryo medium (pH 7.2), and incubated at 28 °C until treatment at 6 h. Only viable eggs were harvested into petri dishes and used for subsequent experiments.

### 2.2. Auranofin (AFN) and Riboceine (RBC) Exposure

The AFN and RBC treatment groups including exposures and doses [17] are outlined in Appendix A. AFN (Sigma-Aldrich, St. Louis, MO, USA, cat. A6733) was made up in Dimethyl Sulfide (DMSO) and kept as a 10 mM stock solution, 100 μM working stocks were made fresh in E3 media for each experiment at a volume of 20 mL. Professor Sally McCormack (University of Otago) gifted Riboceine for these experiments, which was prepared by Chemica Inc. (Los Angeles, CA, USA) and provided by Max International, LLC, Salt Lake City, UT, USA. RBC was diluted in E3 medium at a volume of 20 mL; all the RBC treatments were carried out at the maximal tolerated dose of 500 µM RBC (See Appendix A for RBC alone dose-response). At 6 h post fertilization (hpf), live embryos were randomly distributed into petri dishes (*n* = 30 embryos per dish). The excess media was removed and replaced with the test solution (AFN at 5 µM, AFN at 5 µM with RBC, RBC only) or the untreated control, which contained the equivalent volume of DMSO. For the AFN exposure treatment group, and the AFN and RBC separate treatment group, the initial AFN solution was removed after 24 hpf and replaced with E3 media or RBC, respectively. To assay the phenotypic response to the treatment, 5 dpf larvae were sedated with tricaine and screened for defects using light microscopy; larvae were categorized as (a) normal, (b) having jaw defects or (c) being dead, while the assessor was blinded to the experimental condition the embryos came from. Dead embryos and larvae were excluded from any further analyses.

### 2.3. Cranial Cartilage Measurements

To visualize cartilaginous structures, 5 dpf larvae from the treatment groups were fixed in 4% paraformaldehyde, washed and cleared in methanol. Larvae (*n* = 6 for each treatment group; AFN at 5 µM and AFN at 5 µM with RBC) were stained with Alcian blue solution (0.15% Alcian blue dye in 80:20 EtOH:acetic acid) at 4 °C for 2–3 days. Otago Histology Services Unit (Dunedin, NZ) gifted Alcian Blue (Sigma-Aldrich; cat. A5268). Larvae were rinsed in H_2_O with 0.1% Tween-20 (H_2_O-T) and natural pigments were removed using a bleach solution (0.3% H_2_O_2_ 1% KOH) followed by washes in H_2_O-T. Larvae were incubated for 10 min in 30% saturated borate solution, and subsequently digested with trypsin (Gibco, 0.25% in 30% saturated borate) for 1–2 h at 37 °C. Larvae were washed twice in H_2_O-T and transferred to 80% glycerol for storage. The larvae were imaged using light microscopy in 3% methylcellulose. Cranial cartilage measurements were adapted from those carried out in previous research [28,29]. Measurements of the cranial cartilaginous skeleton were made using the angle and line drawing tools in ImageJ (version 1.48), while the assessor was blinded to the experimental condition the embryos came from.

### 2.4. Detection of Cell Death

Apoptotic cells were measured by whole-mount terminal deoxynucleotidyl transferase-mediated deoxyuridine triphosphate nick-end labeling (TUNEL) using ApopTag Peroxidase in situ Apoptosis Detection Kit (S7100; EMD Millipore Corporation, Billerica, MA, USA) according to the manufacturer’s instructions. A total of 24 hpf embryos were dechorionated with forceps prior to fixing, proteinase K treatment and staining and both 24 hpf and 48 hpf embryos were transferred to 80% glycerol for storage once stained. The embryos were imaged using light microscopy as whole specimens in 3% methylcellulose. Stained cells in the head region were counted manually using ImageJ (version 1.48), while the assessor was blinded to the experimental condition the embryos came from. Two biological replicates were carried out, with 10 embryos per experimental condition (untreated control, RBC only 6 hpf–5 dpf, RBC only 1 dpf–5 dpf, AFN at 5 µM, AFN at 5 µM with RBC concurrent and AFN at 5 µM with RBC separately) in each replicate.

### 2.5. Antioxidant Gene Expression

Embryos from each treatment group (untreated control, AFN at 5 µM, AFN at 5 µM with RBC concurrent and AFN at 5 µM with RBC separately) were collected at 24 hpf and 48 hpf. Zebrafish RNA extracts were prepared from 30 pooled embryos from each treatment group using the NucleoSpin RNA kit (Machery-Nagel, Düren, Germany, cat. 740955.250), according to the manufacturer’s instructions. RNA concentrations and purity were verified on a Nanodrop spectrophotometer. The qScript cDNA SuperMix system (QuantaBio, Beverly, MA, USA, cat. 95048) was used for cDNA synthesis from 1 μg total RNA using oligo dT primers, according to the manufacturer’s instructions. Primers used for quantitative real-time polymerase chain reaction (qRT-PCR) analysis of gene expression have been described previously [17]. SYBR Premix Ex Taq (Tli RNaseH Plus) (Takara Bio, Shiga, Japan, cat. RR820A) was used to amplify cDNA in duplicate with a LightCycler 480 Instrument II (Roche Diagnostics, Indianapolis, USA cat. 05015278001). The cycle threshold (Ct) was calculated by the LightCycler 480 SW1.5 software (Roche Diagnostics, Indianapolis, IN, USA) and normalized to the reference genes (*eif1a* and *rpl13a*) using qBase (Biogazelle, Gent, Belgium). Three biological replicates per treatment were analyzed.

### 2.6. Statistical Analysis

Data were analyzed using the Statistical Package for Social Sciences (SPSS v 22.0, SPSS INC, Chicago, IL, USA). Bivariate analysis was carried out using a one-way analysis of variance (ANOVA). A Tukey post-hoc test was used for multiple comparisons between groups.

## 3. Results

### 3.1. AFN Causes Craniofacial Defects Which Are Rescued by RBC

Consistent with previous reports [17], AFN caused defects of the craniofacial cartilage in 5 dpf zebrafish larvae (Figure 1A,B). In total, 98% of larvae exposed to 5 µM AFN (Figure 1B) had cartilage defects. Concurrent treatment (6 hpf–5 dpf) with 500 µM of RBC (RBC concurrent group) rescued the jaw defects in larvae exposed to 5 µM AFN (Figure 1B). In the concurrent AFN and RBC treatment group, 7% of larvae showed jaw defects (Figure 1B). For reference, 87% of larvae treated with RBC only (6 hpf–5 dpf) had normal cartilage (Figure 1B). To determine whether RBC could rescue the jaw defects after the oxidative stress insult was applied, a treatment group with RBC applied only after larvae had been exposed to 5 µM AFN (RBC separately group) was included. RBC applied separately, after 5 µM AFN, rescued jaw defects, such that 84% of the treated larvae had normal cranial cartilage (Figure 1B).

In larvae exposed to AFN, cranial cartilage defects were assessed by Alcian blue staining (Figure 2A–C) presented as a ventral extension of the ceratohyal and Meckel’s cartilages of the lower jaw; these abnormal jaw morphologies were rescued by concurrent RBC treatment. Exposure to 5 µM AFN both widened the angle of (Figure 2D) and reduced the length of the ceratohyal cartilage (Figure 2E). Phenotypic rescue of the ceratohyal cartilage was observed (Figure 2D,E) in the 5 µM AFN and concurrent RBC group. No differences in the length of the Meckel’s cartilage were observed between the controls, the oxidative stress (5 µM AFN) and rescue groups (AFN and RBC concurrent) (Figure 2F). Exposure to 5 µM AFN increased the distance from the palatoquadrate cartilage of the upper jaw to Meckel’s cartilage of the lower jaw (Figure 2G), which was rescued with concurrent RBC treatment (Figure 2G).

### 3.2. AFN Exposure Results in Apoptosis and Is Rescued by RBC

Exposure to 5 µM AFN resulted in ~6 fold more TUNEL-positive cells than the control group in the developing head at 48 hpf (Figure 3A,B). Treatment with RBC both concurrent with the 5 µM AFN and applied separately, after 5 µM AFN, resulted in a similar number of TUNEL-positive cells in the head, similar to that observed control larvae (Figure 3B).

### 3.3. RBC Treatment Alters Antioxidant Gene Expression

Antioxidant genes glutathione S-transferase P (*gtsp1*), peroxiredoxin-1 (*prdx1*) and glutamate-cysteine ligase (*gclc*) increase in expression in response to oxidative stress caused by AFN exposure in zebrafish embryos [17]. We found no difference in *gclc* expression at 24 hpf (Figure 4A) or 48 hpf (Appendix A) between the treatment groups, with the exception of the AFN- and RBC-treated groups (separately) at 48 hpf. Exposure to 5 µM AFN resulted in higher levels of expression of *gstp1* and *prdx1* in 24 hpf zebrafish embryos (Figure 4B,C). Treatment with RBC alone resulted in no change in the levels of *prdx1 or gstp1,* relative to control embryos (Figure 4B,C). However, concurrent treatment of RBC with AFN resulted in lower levels of *prdx1* and *gstp1* compared to the AFN-treated groups, although the concurrent treatment did not completely rescue expression to levels observed in control embryos (Figure 4B,C). Higher expression of *gstp1* and *prdx1* was observed in AFN-treated embryos at 48 hpf; however, RBC treatment did not rescue the gene expression profiles of *gstp1* and *prdx1* (Appendix A). At 48 hpf (Appendix A), the separate AFN and RBC treatment groups showed *gclc*, *prdx1* and *gstp1* expression levels were similar to the AFN-only treatment. The gene expression analyses indicate that RBC partially (or transiently) rescues AFN-induced antioxidant gene expression changes, and only in the 24 hpf embryos.

## 4. Discussion

In zebrafish, the NCCs migrate into the pharyngeal arches from the respective rhombomeres between 24–48 hpf, and the jaw develops in the hatching period between 48–72 hpf [30]. An increase in oxidative stress during this critical developmental window could affect cell survival of the NCCs [31] and the subsequent development of the cartilaginous precursor cells that contribute to the ossified jaws, with a site specific-effect on jaw morphology and size. We observed cartilaginous jaw defects in 5 dpf zebrafish embryos after treatment with AFN with a ventral extension of the Meckel’s and ceratohyal cartilages consistent with perturbation of NCC development. We show that RBC can also rescue environmental perturbation of craniofacial development induced by AFN. Our TUNEL data (consistent with [21]) indicate that RBC reduces the number of TUNEL-stained cells in the craniofacial region of zebrafish embryos. Our data implicate DNA damage and/or apoptosis of the NCCs as a potential mechanism underlying the craniofacial defects we observed in AFN-treated larvae. Our results are consistent with previous reports indicating that AFN induces defects in NCC-derived cells including the heart and pigmented cells [32]. Surprisingly, we observed rescue of cell death and jaw morphology with RBC treatment even when RBC was applied separately, after AFN treatment. Although RBC is expected to be protective rather than restorative, we suspect that actively growing embryos benefit from RBC’s antioxidant effects after AFN washout, and that RBC may protect from the effects of residual AFN. An increase of GSH synthesis owing to RBC treatment could also enhance recovery from AFN-mediated damage in the growing embryo.

Here, we confirm [17,33] that oxidative stress induced by AFN increases expression levels of the oxidative stress genes *gstp1* and *prxd1* in zebrafish embryos. In humans, a functional polymorphism of *GSTP1* is associated with non-syndromic cleft lip risk, with risk modified in the presence of maternal smoking [34]. Furthermore, in a genetic zebrafish model of Treacher Collins Syndrome, a disorder predominantly characterized by craniofacial abnormalities, the expression of *prxd1* is decreased [35]. The craniofacial defects in this model are rescued by over-expression of Cell nucleic acid binding protein (Cnbp-a ROS-cytoprotective protein) [35] and proteosome inhibition ameliorating proteasome degradation of Cnbp [36], which suggests that these mutants have a cell-redox imbalance. Our data indicate that RBC rescues the expression *gstp1* and *prxd1* and raises the possibility that antioxidant treatment with RBC could ameliorate genetic causes of craniofacial abnormalities by lowering oxidative stress. Moreover, antioxidants, including RBC, have been used previously to rescue craniofacial defects in animal models of genetic craniofacial disorders [6,37,38,39]. In humans, mutations in *SMC1A* are among the cohesin (or cohesin regulator) mutations that cause Cornelia de Lange syndrome (CdLS), a rare multifactorial developmental disorder that includes craniofacial abnormalities [40,41]. In CdLS, mutations in cohesion genes, including *SMC1A,* result in genomic instability, which can lead to delays in DNA repair and downregulation of genes in antioxidant pathways, increasing oxidative stress [42]. RBC was shown to partially rescue craniofacial abnormalities caused by a genetic mutation in *smc1a* in zebrafish [37]. Following on from this, it would be important to investigate if RBC has modifying effects in zebrafish models of other genetic craniofacial disorders including those caused by *GSTP1* [34] and *TCOF1* [6], TXNL4A [43] and DONSON [44], which operate in pathways that could potentially be affected by oxidative stress. In future work, it would be important to define the global transcriptomic outcomes in zebrafish of AFN exposure to identify the genes and pathways that are rescued by RBC. This information would likely illuminate the mechanisms by which RBC rescues DNA damage and/or apoptosis.

It is important to note that ROS are crucial in normal embryonic development driving normal cell proliferation and differentiation [9,45] and that antioxidant treatment can also be toxic [46]. NCCs, in particular, are extremely sensitive to any perturbation of ROS given their increased oxidative state and contribute to many cellular fates, including chondrocytes, melanocytes, neurons and cardiac cells [47]; thus, systemic treatment with RBC and reduction of ROS might be detrimental in certain cellular contexts that were not assessed here. Compared with controls, there was a slight increase in the incidence of developmental defects upon treatment with the 500 uM RBC dose that rescued AFN-induced craniofacial abnormalities. However, we have not investigated the efficacy of lower RBC doses and milder circumstances of oxidative stress that are likely to more closely reflect environmental conditions. Finally, whether the results from our study would translate to human embryogenesis is very much unknown [48].

## 5. Conclusions

Our data indicate that the antioxidant RBC rescues AFN-induced craniofacial abnormalities in the zebrafish, likely by countering oxidative stress. Thus, RBC could be useful in the prevention of craniofacial defects owing to environmental or genetic risk.

## Figures and Tables

**Figure 1 antioxidants-10-01964-f001:**
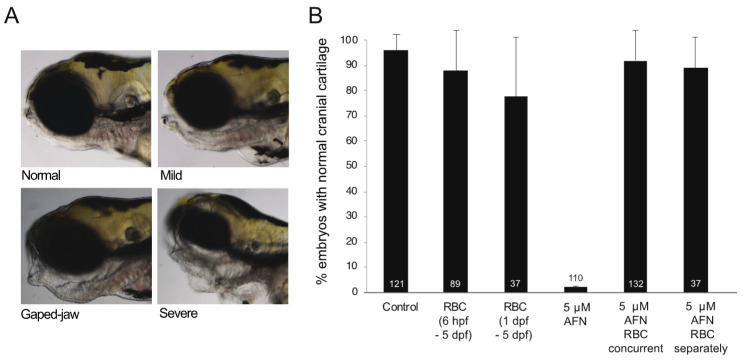
AFN causes craniofacial defects which are rescued by RBC. (**A**) Representative images of larvae at 5 dpf (lateral view) with normal cartilage (**top left**) and abnormal cartilage; mild (**top right**), gaped jaw (**bottom left**) and severe (**bottom right**) as a result of 5 μM AFN treatment. (**B**) Proportion of larvae with normal craniofacial cartilage in untreated controls, RBC (500 µM) exposure 6 hpf–5 dpf, RBC (500 µM) exposure 1–5 dpf, 5 µM AFN only, 5 µM AFN with concurrent exposure to RBC (500 µM) starting at 6 hpf through to 5 dpf, 5 µM AFN with separate exposure to RBC (500 µM) with RBC treatment commencing at 1 dpf through to 5 dpf. Error bars show the standard error of the mean from three biological replicates. Data labels show the total number of larvae observed for each treatment group. Auranofin, AFN; Riboceine, RBC; hours post fertilization, hpf; days post fertilization, dpf.

**Figure 2 antioxidants-10-01964-f002:**
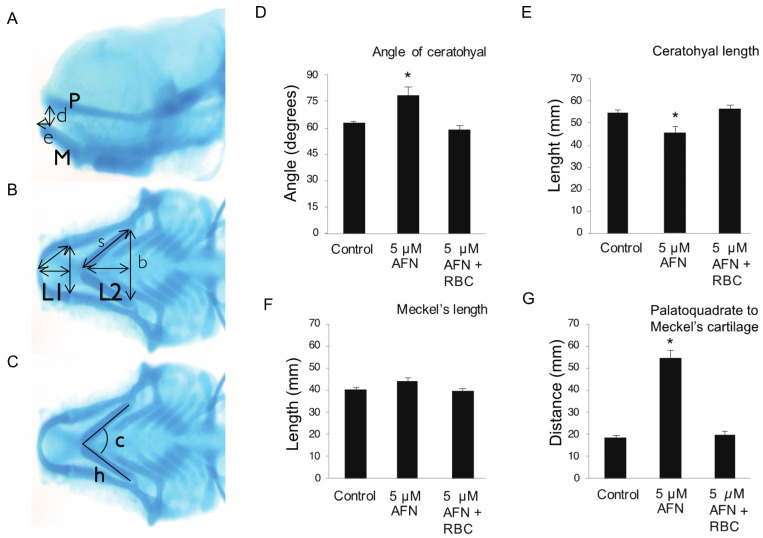
Measurements of cartilaginous structures of the zebrafish jaw as assessed by Alcian blue staining. (**A**) Sagittal view of the head: P—palatoquadrate cartilage, M—Meckels cartilage, d—distance (mm) from P to M, e—extension of M past P. (**B**) Ventral view of the head: L1—length of Meckels cartilage (mm), L2—Length of ceratohyal cartilage (mm), length of the cartilages was estimated using the Pythagorean formula L = √ (s2 − b2/4). (**C**) Ventral view of the head: ch—angle of the ceratohyal cartilage. Data for angle of ceratohyal (degrees) (**D**), ceratohyal length (mm) (**E**), Meckel’s length (mm) (**F**) and distance from palatoquadrate to Meckel’s cartilage (mm) (**G**). Treatment groups; untreated controls, 5 µM AFN and 5 µM AFN with concurrent RBC exposure (6 hpf–5 dpf). Error bars show the standard error of the mean. Mean is the average of six larvae for each treatment group. Asterisk indicates values that are statistically significantly different (*p* < 0.01) as assessed by one-way ANOVA, Tukey post-hoc analysis. All linear measurements are in mm. Auranofin, AFN; Riboceine, RBC; hours post fertilization, hpf; days post fertilization, dpf.

**Figure 3 antioxidants-10-01964-f003:**
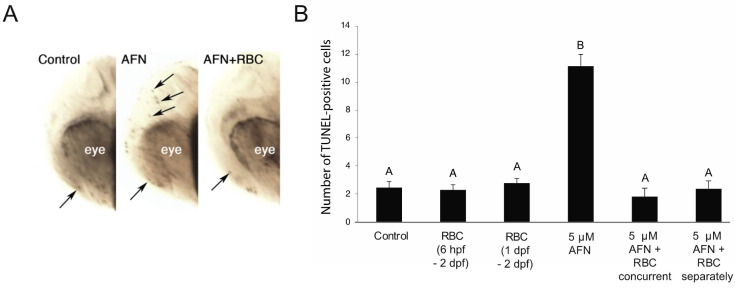
AFN exposure results in apoptosis and rescued by RBC. (**A**) Images of TUNEL-stained apoptotic cells in the head of embryos at 24 hpf (lateral view) in untreated controls (left) and 5 µM AFN-treated (middle) and RBC + AFN concurrently treated (right) embryos. Arrows indicate TUNEL-positive cells. (**B**) Number of TUNEL-positive cells in the head of 48 hpf embryos. Treatment groups: untreated controls, RBC (500 µM) exposure starting at 6 hpf, RBC (500 µM) exposure starting at 1 dpf, 5 µM AFN, 5 µM AFN with concurrent RBC exposure (6 hpf–5 dpf) and 5 µM AFN with separate RBC exposure (1 dpf–5 dpf). Error bars are standard error of the mean. Mean is the average of two biological replicates with *n* = 10 embryos in each treatment group. Data not sharing the same letter are statistically significantly different (*p* < 0.01) as assessed by one-way ANOVA, Tukey post-hoc analysis. Auranofin, AFN; Riboceine, RBC; hours post fertilization, hpf; days post fertilization, dpf.

**Figure 4 antioxidants-10-01964-f004:**
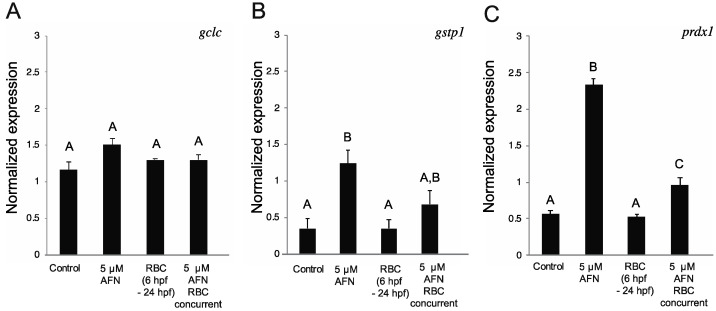
RBC treatment alters AFN-induced antioxidant gene expression response. The normalized expression levels of *gclc*; (**A**) *gstp1*; (**B**) and *prdx1*; (**C**) from untreated controls, RBC (500 µM) exposure (6 hpf–5 dpf), 5 µM AFN at 24 hpf and 5 µM AFN with concurrent RBC exposure (starting at 6 hpf). Error bars are standard error of the mean from three biological replicates (*n* = 10 embryos in each replicate). Data not sharing a letter are statistically significantly different (*p* < 0.05) as assessed by one-way ANOVA, Tukey post-hoc analysis. Auranofin, AFN; Riboceine, RBC; hours post fertilization, hpf; days post fertilization, dpf.

## Data Availability

The data presented in this study are available in this manuscript and Appendix A.

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
