# Peer review of "Riboceine Rescues Auranofin-Induced Craniofacial Defects in Zebrafish"

_antioxidants, 2021, doi:10.3390/antiox10121964_

Round 1

Reviewer 1 Report

The authors presented a study entitle “Riboceine Rescues Auranofin-Induced Craniofacial Defects in Zebrafish”. The main objective of this study was to evaluate the ability of Riboceine (RBC) in rescuing craniofacial defects arising from exposure to auranofin (AFN).

General comments:

Introduction:

The objective of this investigation is interesting, but the information is very scarce, there is no background on the importance of antioxidants in ensuring correct embryonic development. Above all, there is no literature review regarding the already known effects of RBC and the right motivation that led the authors of this study to choose this specific antioxidant rather than another. In my opinion, the introduction should be revised by adding more references to support the objective.

Results and Discussion:

The interesting data that emerges from this study is that RBC is able to restore defects already established as can be seen from the separate treatment data.

This is an important finding as antioxidants are known to work by preventing oxidative damage rather than restoring damage. This aspect has not been treated with due importance. The authors were too elusive and unclear.

In general, discussion section, as for the introduction, lacks supporting data and not focus results well.

Furthermore, authors need to include the main limitations of this study, particularly related to the double face of antioxidants, which can become toxic at high concentrations, and the difficulty of translating the results obtained on animal models to humans; I suggest some useful references to the authors:

https://doi.org/10.3390/antiox10071137; 10.1016/j.ecoenv.2020.110642; https://doi.org/10.3390/ijms13044655

Future perspectives need to be included: what other gene expression pathways can be investigated? are the effects of RBC on DNA Damage Response pathways known?

Conclusion:

the conclusions reached by the authors are too speculative.

Specific comment:

In the Material and Methods section, the chemical list paragraph with the brand of the chemicals and where to buy it should be added.

Also give more details on the breeding of zebrafish and the conditions for obtaining the embryos.

In results section the authors rightly exclude dead embryos from future analyzes. However, it would be interesting to know the survival rate of the embryos following each treatments.

Figure 1A is unclear and misleading, it would be better to eliminate it.

A complete revision of the text is recommended and beware of typing errors. 

Reviewer 2 Report

Antioxidants MS ID1477808

Leask et al have investigated the interaction between oxidative stress chemicals (e.g., Auranofin, AFN) on craniofacial abnormalities using the zebrafish model. As a result, they found the Reboceine (RBC) rescued the developmental defects such as craniofacial defects focusing on risk genes which are associated with cranial cartilage deficits, in particular morphological distortion of Mackel’s cartilage, reduce cell apoptosis utilizing quantitative analysis of antioxidant gene expression, cell death, cranial cartilage measurement with imaging. Birth defects are a burden for children's health and raise the pressing issue of health economics. It remains the risk as causative for Craniofacial birth defects due to a variety of environmental stressor cause risk with potential consequence of gene-environment byproduct.

Study design is well constructed, and data interpretation is reasonable to explore impact of RBC on AFN driven craniofacial defects as a model of developmental disorder with underlying cellular and molecular mechanisms using in vitro models like zebrafish rather than mouse model or cellular model. This study examines the effect of D-Ribose-L-Cysteine (RiboCeine), a kind of dietary supplement on AFN mediated craniofacial defects using zebrafish. 

The length of the article is adequate, neatly documented and well justified using their strategy like exploring networks between molecular and cellular to promote mechanistic study on the effect of RBC on AFN as models of craniofacial defects use zebrafish.  However, I’d like to discuss there is some degree of shortage (? ) or justification for inquiries raised : role of those three genes of antioxidant in the development period, molecule alteration following environmental factors such as epigenetic, and functionality assay to explain connectivity of development defect focusing on three genes and its mapping to cell types.

Minor issues,

To validate the oxidative stress specific or in general, I was wondering if the authors examined the difference effect of NAC and RBC regarding the pattern of antioxidant genes they found in 48 hdf. Have they observe difference MnSOD and other redox enzymes between AFN and AFN+RBC in embryo and zebrafish?

In fig 2, is there any evidence to support the impact of RBC to palatoquandrate of meckel’s cartiliage following  AFN exposure regarding gene alteration such as gclc, gstp1, or prdx1?

In Fig 3, is there any evidence of TUNNEL as a result referring to AFN +RBC?

In connect to supplemental data at Figure 2, authors have any evidence to support the rescue cranial cartilage defect as morphological changes either embryo or staining of dual bone and cartilage along with three gene expression level in antioxidant genes expression in Supple Fig 2 or any gene knockout experiments using zebrafish model? Otherwise, have been examined the protein expression level to confirm potential risk of antioxidant gene expression

Is there any exome sequence they conducted or not?

In a previous study, TXNL4A pathogenic variants were reported in the Burn-McKeown Syndrome (BMKS). So, have you measured effect of AFN and AFN+RBC regarding change of expression (or staining) level either gene or Thioredoxin-interacting protein (TXNIP) using embryo

Consider the restructure of title, for example, Riboceine rescues Auranofin-induced craniofacial defects in zebrafish: potential gene risk and craniofacial cartilage defects

Round 2

Reviewer 1 Report

The authors made the suggested changes clearly and precise. In my opinion, the manuscript has certainly improved and it should be accepted in the present form. 

This manuscript is a resubmission of an earlier submission. The following is a list of the peer review reports and author responses from that submission.